# Functional Analysis of Direct In Vitro Effect of Phosphorylated Tau on Mitochondrial Respiration and Hydrogen Peroxide Production

**DOI:** 10.3390/biom15040495

**Published:** 2025-03-28

**Authors:** Zdeněk Fišar, Jana Hroudová

**Affiliations:** 1Department of Psychiatry, First Faculty of Medicine, Charles University and General University Hospital in Prague, Ke Karlovu 11, 120 00 Prague, Czech Republic; hroudova.jana@gmail.com; 2Department of Pharmacology, First Faculty of Medicine, Charles University and General University Hospital in Prague, 120 00 Prague, Czech Republic

**Keywords:** Alzheimer’s disease, mitochondrial dysfunction, phosphorylated tau, respiratory state, hydrogen peroxide, isolated mitochondria

## Abstract

The neurotoxicity of phosphorylated tau protein (P-tau) and mitochondrial dysfunction play a significant role in the pathophysiology of Alzheimer’s disease (AD). In vitro studies of the effects of P-tau oligomers on mitochondrial bioenergetics and reactive oxygen species production will allow us to evaluate the direct influence of P-tau on mitochondrial function. We measured the in vitro effect of P-tau oligomers on oxygen consumption and hydrogen peroxide production in isolated brain mitochondria. An appropriate combination of specific substrates and inhibitors of the phosphorylation pathway enabled the measurement and functional analysis of the effect of P-tau on mitochondrial respiration in defined coupling control states achieved in complex I-, II-, and I&II-linked electron transfer pathways. At submicromolar P-tau concentrations, we found no significant effect of P-tau on either mitochondrial respiration or hydrogen peroxide production in different respiratory states. The titration of P-tau showed a nonsignificant dose-dependent decrease in hydrogen peroxide production for complex I- and I&II-linked pathways. An insignificant in vitro effect of P-tau oligomers on both mitochondrial respiration and hydrogen peroxide production indicates that P-tau-induced mitochondrial dysfunction in AD is not due to direct effects of P-tau on the efficiency of the electron transport chain and on the production of reactive oxygen species.

## 1. Introduction

### 1.1. Mitochondrial Dysfunction in Alzheimer’s Disease

Mitochondrial dysfunction is associated with aging and with neurodegenerative diseases such as Alzheimer’s disease (AD) [1,2,3]. AD is a progressive neurodegenerative disease that results from neuronal death and the loss of synapses in specific brain regions. Neuropathologically, AD is characterized by an increased occurrence of extracellular neuritic plaques composed of amyloid beta (Aβ) and intracellular neurofibrillary tangles (NFTs) composed of paired helical filaments of hyperphosphorylated tau protein (P-tau) [4,5]. At the biochemical level, AD is considered an Aβ amyloidosis [6] and a secondary tauopathy [7]. There is evidence that soluble Aβ and tau oligomers are responsible for brain cell damage and death in AD [8], with neuroinflammation [9] and oxidative stress [10,11] being the main contributors to the pathophysiology of AD. It is believed that the neurotoxicity of soluble tau oligomers is involved in the progression of neurodegeneration in the prodromal period and in AD dementia [12,13,14]. The target of Aβ and P-tau in AD appears to be primarily synaptic mitochondria [15].

Alzheimer’s amyloidopathy and tauopathy are associated with neurodegeneration through mitochondrial dysfunction and abnormal mitochondrial morphology and dynamics [16,17,18,19]. According to the mitochondrial cascade hypothesis of sporadic AD [20], mitochondrial dysfunction may be a trigger for the development of AD. Attention is paid primarily to clarifying the amyloid, tau, neurotransmitter, metabolic, and mitochondrial hypotheses of AD [21,22,23] and the interconnection of these hypotheses [24].

Impaired bioenergetics and increased oxidative stress are essential in the neurodegenerative processes in AD. The overproduction of reactive oxygen species (ROS) plays a crucial role in both Aβ pathology [25], P-tau pathology [12], and mitochondrial dysfunction [2] in AD. It is not yet clear whether mitochondrial dysfunction in AD triggers Aβ and P-tau pathology or whether it is a consequence of them.

### 1.2. Tau Neurotoxicity

The microtubule-associated protein tau stabilizes microtubules; in addition, tau is involved in the regulation of axonal transport, DNA integrity, actin filament formation, and glutamate NMDA receptor signaling [26]. There are six isoforms of tau protein in the brain, ranging in length from 352 to 441 amino acids. Tau phosphorylation is controlled by the activities of protein kinases and phosphatases [27]. The kinases glycogen synthase kinase-3beta (GSK-3) and cyclin-dependent kinase 5 are mainly involved in tau hyperphosphorylation in AD [28,29].

The tau hypothesis of AD is based on the observation that NFTs are composed of P-tau [30] and postulates that the primary cause of AD development is microtubule destabilization and the neurotoxicity of P-tau and its aggregates [31,32].

P-tau monomers are misfolded and give rise to detergent soluble P-tau oligomers and detergent insoluble granular tau oligomers, which can further form paired helical filaments and NFTs [33]. P-tau neurotoxicity is caused by soluble P-tau oligomers rather than by NFTs [34,35,36,37]. P-tau oligomers are thought to be involved in the disruption of synaptic and mitochondrial function, including mitochondrial bioenergetic, dynamics, transport, and mitophagy [38]. Their neurotoxic effects include changes in membrane permeability and the resulting ion imbalance [39]. Once small amounts of tau inclusions form, they can self-propagate in a prion-like manner [40]. The way tau pathology spreads and its connection to functional connectivity is still a subject of research [41].

Aβ and P-tau may act synergistically, as Aβ oligomers trigger the formation of P-tau oligomers and P-tau oligomers increase the toxicity of Aβ oligomers [42].

### 1.3. Mitochondrial Toxicity of Tau

Extracellularly applied tau oligomers are internalized by cells [43], and some tau is located on the outer mitochondrial membrane and in the inner mitochondrial space [44]. This allows for a tau-dependent regulation of mitochondrial functions [45] through direct interactions with mitochondrial proteins and membranes. Tau shows a high affinity for membranes enriched with cardiolipin, which plays a significant role in mitochondrial membrane stability and supercomplex formation and thus in the bioenergetic efficiency of the electron transfer system (ETS) [46,47]. The effects of P-tau on mitochondrial function may also be indirect, through alterations in signaling pathways associated with mitochondrial function that leads to the disruption of mitochondrial transport and dynamics [48].

Tau impairs mitochondrial bioenergetics, dynamics, transport, mitophagy, neurosteroidogenesis, calcium efflux, and the coupling of mitochondria with the endoplasmic reticulum [38,45,49,50]. Evidence for mitochondrial function impairment by P-tau aggregates comes primarily from in vitro studies using cell lines or from animal studies.

In a transgenic mouse model of AD, the synergistic effects of Aβ and tau on reducing membrane potential and ATP production were confirmed. The reduction in complex I activity was tau-dependent, while the reduction in complex IV activity was Aβ-dependent. Higher levels of ROS were observed in old transgenic mice [51,52].

The overexpression of tau in the SH-SY5Y cell line led to reduced complex I activity and reduced ATP levels, as well as increased ROS production [53]. The bioenergetic deficit induced by P-tau in SH-SY5Y was manifested in reduced maximal respiration and respiratory spare capacity [54] and decreased ATP synthesis and mitochondrial membrane potential [55]. P-tau reduces complex I and V expression, complex I activity, and ATP production, decreases calcium buffering capacity, lowers mitochondrial membrane potential, increases ROS production, and affects mitochondrial permeability transition pore (mPTP) formation [15,56,57]. The molecular mechanisms of these effects of P-tau on mitochondrial function remain to be elucidated.

### 1.4. Aim

The cause of P-tau neurotoxicity in AD may be damage to synapses and brain cells due to impaired bioenergetics and oxidative stress, i.e., a reduced efficiency of mitochondrial oxidative phosphorylation and an increased production of ROS by mitochondria [58]. Mitochondrial localizations of tau and tau-induced mitochondrial dysfunction have been well established [44,47,48,56,57,59,60]. The role of direct interactions of P-tau oligomers with mitochondria therefore needs to be studied.

To uncover the mechanisms leading to P-tau-induced mitochondrial toxicity, we studied the direct effects of P-tau on the kinetics of oxygen consumption and the associated H_2_O_2_ production in defined mitochondrial respiratory states. We performed measurements on a biological model of isolated brain mitochondria, which allows us to determine in which respiratory states these changes occur. The aim was to determine whether the direct interaction of GSK-3 phosphorylated P-tau oligomers with mitochondrial membranes and proteins leads to mitochondrial dysfunction.

## 2. Materials and Methods

Simultaneous P-tau-induced changes in mitochondrial respiration (polarographic measurements) and H_2_O_2_ production (fluorometric measurements) were measured in isolated mitochondria using the Oxygraph-2k equipped with smart Fluo-Sensors (Oroboros Instruments, Innsbruck, Austria).

### 2.1. Media and Chemicals

Mitochondrial isolation medium: 0.32 mol/L sucrose buffered with 4 mmol/L HEPES (pH 7.4).

Mitochondrial respiration medium (MiR05 without BSA): 110 mmol/L sucrose, 60 mmol/L K^+^-lactobionate, 20 mmol/L taurine, 3 mmol/L MgCl_2∙_6H_2_O, 10 mmol/L KH_2_PO_4_, 0.5 mmol/L EGTA, and 20 mmol/L HEPES (pH 7.1) [61].

Mitochondrial preservation medium: MiR05 supplemented with 20 mmol/L histidine, 20 µmol/L vitamin E succinate, 3.0 mmol/L reduced L-glutathione, 1.0 µmol/L leupeptine, 2 mmol/L glutamate, 2 mmol/L malate, and 2 mmol/L Mg-ATP [62].

The stock solutions: 0.8 mol/L malate, 2 mol/L pyruvate, 2 mol/L glutamate, 1 mol/L succinate, 0.5 mol/L ADP in 0.3 mol/L MgCl_2_, 1 mmol/L rotenone in ethanol, 1 mmol/L carbonyl cyanide-*p*-trifluoromethoxyphenylhydrazone (FCCP), 0.5 mg/mL antimycin A in ethanol, 10 mmol/L Na_2_HPO_4_ (NaP, pH 7.4), 500 µmol/L tris(2-carboxyethyl)phosphine (TCEP) in NaP, 0.78 mmol/L *N*-ethylmaleimide (NEM) in NaP, and 8.455 µmol/L Tau-441.

BLANK solution: 37.5 µmol/L TCEP and 30 µmol/L NEM in saline with TRIS.

Chemicals were purchased from Merck (Merck KGaA, Darmstadt, Germany), except for (1) Amplex™ UltraRed from Molecular Probes (Molecular Probes INC, Eugene, OR 97402, USA) and (2) P-tau (TAU-H5147 Human Tau-441 GSK-3beta-phosphorylated protein) from ACROBiosystems (ACROBiosystems Group, Newark, DE 19711, USA); P-tau was supplied and shipped with dry ice in buffer 50 mmol/L Tris, 150 mmol/L NaCl, pH 7.5, with glycerol as a protectant.

### 2.2. Mitochondria Isolation

Mitochondria were isolated from the cerebral cortex of pig brain by a previously described method [63]. Pig brains were obtained from a local slaughterhouse within 1 h after CO_2_ stunning of animals and killing them via bleeding them to death. Briefly, the brain cortex was gently homogenized in ten volumes (*w*/*v*) of ice-cold buffered sucrose 0.32 M supplemented with a protease inhibitor cocktail. The crude mitochondrial fraction was prepared by differential centrifugation, and then the crude mitochondrial fraction was purified by density-gradient centrifugation on a sucrose gradient. Purified mitochondria were used to eliminate interactions of P-tau with residues of other organelles contaminating the crude mitochondrial fraction. Purified mitochondria were stored at 2 °C (in an ice-cold water bath) in preservation medium at a protein concentration of approximately 40 mg/mL determined by the Lowry method. In previous experiments, we verified that purified mitochondria prepared in this way are a suitable biological model for the study of drug-induced mitochondrial dysfunction [64].

### 2.3. P-Tau Oligomer Preparation

P-tau oligomers were prepared by a simplified validated method according to [43]. Briefly, 50 µg P-tau-441 in 123.4 µL buffer was added to 37.5 µmol/L TCEP (5 × molar excess for 7.5 µmol/L Tau-441). This was incubated for 1 h at 4 °C. Then, 30 µmol/L NEM (4 × molar excess for 7.5 µmol/L Tau-441) was added. The resulting solution was 7.50 µmol/L Tau-441 in medium with 37.5 µmol/L TCEP and 30.3 µmol/L NEM. This was incubated at 4 °C overnight.

### 2.4. Mitochondrial Respiration and Hydrogen Peroxide Production

The in vitro effect of P-tau on mitochondrial respiration and hydrogen peroxide production was measured using an O2k respirometer with O2k-Fluo Smart-Module (Oroboros Instruments Corp, Innsbruck, Austria), allowing for the simultaneous electrochemical measurement of oxygen consumption (Clark electrode) together with the fluorometric monitoring of H_2_O_2_ production [65].

The measurement of oxygen consumption was provided by Clark electrodes. The measurement of H_2_O_2_ released from isolated mitochondria was provided by smart fluo-sensors with green emission (LED 525 nm) and photodiodes, which were connected to the measurement chambers and measured the red fluorescence of UltroxRed (like resorufin), a product of the reaction of H_2_O_2_ and Amplex™ UltraRed catalyzed by horseradish peroxidase (HRP). Samples were measured simultaneously in two 2 mL chambers (stirred, at 37 °C), where after oxygen saturation, the chambers were closed and the oxygen flux per mass (nmol O_2_ per sec per mg protein) and the kinetics of H_2_O_2_ production (nmol H_2_O_2_ per sec per mg protein) were measured simultaneously.

Validated substrate–inhibitor–titration protocols were used to establish the P-tau-induced changes in the oxygen consumption and H_2_O_2_ production of the isolated mitochondria [65,66,67,68,69]. In isolated brain mitochondria, we measured (1) the effect of different concentrations of P-tau (3 to 120 nmol/L) on the respiration and H_2_O_2_ production in the complex I-, II-, and I&II-linked pathways and (2) 60 nmol /L P-tau effect on respiration and H_2_O_2_ production in various coupling control states of the complex I&II-linked pathway.

The measurement protocol was used according to [65] as follows:

To 2 mL of oxygen-saturated MiR05-BSA, add 10 µmol/L Amplex UltraRed, 1 U/mL HRP, 5 U/mL superoxide dismutase, and mitochondria at a final concentration of 0.1–0.2 mg/mL. For the calibration of hydrogen peroxide production, add 0.2 µmol/L H_2_O_2_.

(1) To activate the CI-linked pathway and measure the effect of different concentrations of P-tau, add 2 mmol/L malate, 5 mmol/L pyruvate, 10 mmol/L glutamate, and 1.25 mmol/L ADP in sequence. Then, titrate in one measuring chamber with 3.75, 7.5, 15, 30, 60, and 120 nmol/L P-tau and in the other chamber with equal volumes of BLANK solution. Finally, add 20 ng/mL oligomycin, 1–2 µmol/L FCCP, and 1 µmol/L rotenone.

(2) To activate the CII-linked pathway and titrate P-tau, add 1.25 mmol/L ADP, 10 mmol/L succinate, and 1 µmol/L rotenone in sequence. Then, titrate in one measuring chamber with 3.75, 7.5, 15, 30, 60, and 120 nmol/L P-tau and in the other chamber with equal volumes of BLANK solution. Finally, add 20 ng/mL oligomycin, 1–2 µmol/L FCCP, and 1.25 µg/mL antimycin A.

(3) To activate the CI&II-linked pathway and titrate P-tau, add 2 mmol/L malate, 5 mmol/L pyruvate, 10 mmol/L glutamate, 1.25 mmol/L ADP, 10 mmol/L succinate, and 1 µmol/L rotenone in sequence. Then, titrate in one measuring chamber with 3.75, 7.5, 15, 30, 60, and 120 nmol/L P-tau and in the other chamber with equal volumes of BLANK solution. Finally, add 20 ng/mL oligomycin, 1–2 µmol/L FCCP, and 1.25 µg/mL antimycin A.

(4) To measure the effect of P-tau on oxygen consumption and H_2_O_2_ production in different respiratory states for the CI&II-linked pathway, add 10 µmol/L Amplex UltraRed, 1 U/mL HRP, and 5 U/mL superoxide dismutase to 2 mL of MiR05-BSA. Then, add 60 nmol/L P-tau to one measuring chamber and the same volume of BLANK to the other. Gradually add mitochondria at a final concentration of 0.1–0.2 mg/mL, 0.2 µmol/L H_2_O_2_ (for the calibration of hydrogen peroxide production), 2 mmol/L malate, 5 mmol/L pyruvate, 10 mmol/L glutamate, 1.25 mmol/L ADP, 10 mmol/L succinate, 1 µmol/L rotenone, 20 ng/mL oligomycin, 1–2 µmol/L FCCP, and 1.25 µg/mL antimycin A.

### 2.5. Data Analysis

DatLab software version 7.4.0.4 (Oroboros Instruments Corp, Innsbruck, Austria) was used for respirometry and fluorometry data acquisition and analysis. To correct for a slight changes in the signals during experiments lasting about 1 h, the relative respiratory rate and the relative rate of H_2_O_2_ production were calculated as the ratio of the oxygen flux or rate of H_2_O_2_ formation in the sample with P-tau to the same parameters in the sample with BLANK.

Statistical analyses were performed using a TIBCO Statistica 14.0.0.15 data analysis software system (TIBCO Software Inc., Palo Alto, CA, USA). Data are presented as the mean ± standard deviation (SD).

## 3. Results

P-tau oligomers were screened to determine the effects of the P-tau on mitochondrial respiration and hydrogen peroxide production by isolated purified mitochondria.

We did not find a statistically significant effect of P-tau oligomers on oxygen flux and H_2_O_2_ production in different respiratory states in the complex I-linked (NADH pathway), complex II-linked (succinate pathway), or in complex I & II-linked pathway.

We found only insignificant changes in oxygen flux and H_2_O_2_ production with increasing P-tau concentration in the sample (Figure 1).

The simultaneous measurement of the kinetics of oxygen consumption and H_2_O_2_ production in defined respiratory states showed almost identical results (Figure 2) after the addition of 60 nmol/L P-tau and after the addition of medium without P-tau (BLANK).

## 4. Discussion

Aβ pathology, P-tau pathology, oxidative stress, and neurotransmitter imbalance are interlinked pathogenic mechanisms of AD [70]. Mitochondrial dysfunction is involved in all these pathologies. It is not yet clear which process (biomarker) triggers the development of AD (years before the onset of clinically detectable symptoms of the disease), nor whether such a trigger is common to the entire group of sporadic AD. According to the integrated AD hypothesis, the mutual interaction and synergy of Aβ, P-tau, and mitochondria are responsible for the development of AD.

In this study, we performed a functional analysis of the direct in vitro effect of P-tau on mitochondrial respiration and hydrogen peroxide production. Based on tau-induced mitochondrial dysfunction observed both in cell lines and in animal studies (reduced complex I activity and increased ROS production), we expected a reduction in mitochondrial respiration, especially in the NADH pathway. We did not confirm the hypothesis that P-tau oligomers have a direct effect on mitochondrial oxygen consumption and hydrogen peroxide production by mitochondria in any defined respiratory state. There is a need to study synergic effects of Aβ and tau on mitochondrial functions.

Our results indicate that P-tau-induced mitochondrial dysfunction is caused by indirect effects, such as impaired mitochondrial transport due to microtubule disruption and altered mitochondrial dynamics [71], or the binding of P-tau to voltage-dependent anion-selective channel 1 (VDAC-1) in the outer mitochondrial membrane [72] rather than directly affecting ETS efficiency.

Changes in mitochondrial functions measured in vitro using a model of isolated brain mitochondria may not fully correspond to changes in whole cells due to the absence of interactions with cytosolic components and other organelles. The risk has been minimized by using proven solutions for isolating and storing brain mitochondria and by using well-established measurement protocols for the measurement of oxygen consumption kinetics and H_2_O_2_ production [63,65]. Isolated mitochondria provide the ability to sensitively measure these mitochondrial functions in defined respiratory states and interpret the results in terms of changes in complex I-linked and complex II-linked pathways. The disadvantage of the used model can be considered the impossibility or unsuitability of measuring fission and fusion, biogenesis, autophagy, mitophagy, and mitochondrial functions associated with the endoplasmic reticulum or with bioenergetic processes taking place in the cytosol.

These findings suggest that although P-tau is sufficient to induce neurodegeneration [73], it does not occur through rapid direct interactions with the mitochondrial ETS. In this sense, our study contributed to the understanding of the molecular mechanisms of the mitochondrial toxicity of P-tau oligomers in AD and thereby to the identification of suitable mitochondrial targets for new AD drugs.

The biggest challenge in the development of new AD drugs is to select the most effective and selective agents with respect to their safety and potential toxic side effects. To achieve this, it is necessary to target neurodegenerative processes, especially Aβ and P-tau pathology and the associated mitochondrial dysfunction. New AD drugs target the processes of Aβ and P-tau neurotoxicity, mitochondrial dysfunction, oxidative stress, metabolic disorders, and neuroinflammation [70,74,75,76,77,78].

The symptomatic treatment of AD is aimed at regulating brain neurochemistry. Targets for new causal and drug-modifying drugs are sought primarily in the signaling pathways of the neurotoxicity of Aβ and P-tau oligomers [79,80] and in the regulation of mitochondrial functions related to this pathology. For the restoration of mitochondrial function impaired in AD, regulating the efficiency of the oxidative phosphorylation system [81] and eliminating mitochondrial toxicity of both Aβ and P-tau is a promising goal.

Current anti-tau therapies (e.g., GSK-3 inhibitors or tau aggregation inhibitors) are not fully successful and are under investigation [82,83]. Promising therapeutic targets for AD may include protein kinases and phosphatases modulating tau phosphorylation, tau acetylation inhibitors, tau glycosylation modulators, microtubule stabilizers, tau truncation modulators, and tau aggregation inhibitors [82,84,85]. Mitochondrial dysfunction is a key process associated with neurodegeneration and therefore, a promising therapeutic target for new causal drugs against AD is to eliminate both the formation and mitochondrial toxicity of Aβ and tau oligomers [56,86]. Eliminating the neurotoxicity of P-tau oligomers appears to be a suitable target for new anti-AD drugs in the prodromal period of AD. Measuring the effect of new AD drugs on tauopathy in AD, especially those substances that prevent tau aggregation, appears promising [82,84,87].

Finding new AD drugs requires a better understanding of the pathology of amyloid beta and P-tau in relation to mitochondrial dysfunction. The regulation of mitochondrial function is a suitable therapeutic strategy for metabolic and neurodegenerative diseases, especially those with age as a risk factor [88,89,90]. Understanding the effects of P-tau oligomers on mitochondrial function will help to find the most effective agents in terms of eliminating P-tau neurotoxicity in AD and other tauopathies [70].

## 5. Conclusions

The hyperphosphorylation of tau and its abnormal accumulation and aggregation are involved in the neurodegenerative processes in AD. The exact mechanism of P-tau regulation in the diseased conditions is unclear. Mitochondrial dysfunction is involved in P-tau pathology; therefore, both the in vitro and in vivo effects of P-tau on mitochondrial function, such as oxygen consumption or hydrogen peroxide production, are studied. The dysfunction of these parameters can be attributed to impaired bioenergetics and increased oxidative stress in AD.

We did not find a significant functional alteration of mitochondrial respiration and H_2_O_2_ production induced by the direct interaction of P-tau oligomers with isolated mitochondria. This indicates that the mitochondrial toxicity of P-tau observed in cell line studies and in animal studies is not caused by the direct effects of P-tau on mitochondrial ETS efficiency and the associated hydrogen peroxide production.

Tau pathology and mitochondrial dysfunction are attractive targets for the treatment of AD and other tauopathies. Since there is no significant direct effect of P-tau oligomers on mitochondrial respiration and H_2_O_2_ production, it seems more appropriate to target new AD drugs to inhibit the formation and spread of P-tau oligomers rather than their direct interactions with mitochondrial proteins. However, the synergy of the direct effects of Aβ and P-tau on mitochondrial function remains to be tested. In view of the effects of P-tau observed in animal and cell line studies, the administration of agents that increase/restore mitochondrial bioenergetics appears to be suitable for eliminating P-tau mitochondrial toxicity but not as a causal treatment to prevent the development of P-tau-induced mitochondrial dysfunction.

## Figures and Tables

**Figure 1 biomolecules-15-00495-f001:**
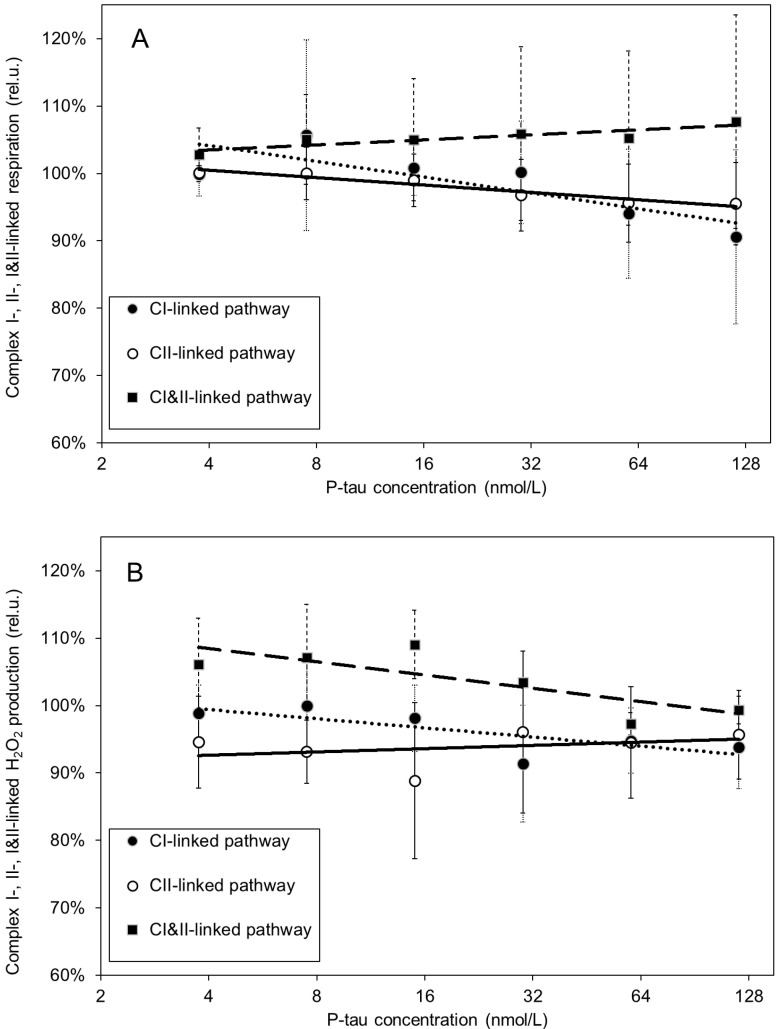
In vitro effect of the phosphorylated tau (P-tau) on (**A**) mitochondrial respiration and (**B**) hydrogen peroxide production, both for complex I-, complex II-, and complex I&II-linked electron-transfer pathways in isolated mitochondria. Percent of control is expressed as the mean ± SD (%).

**Figure 2 biomolecules-15-00495-f002:**
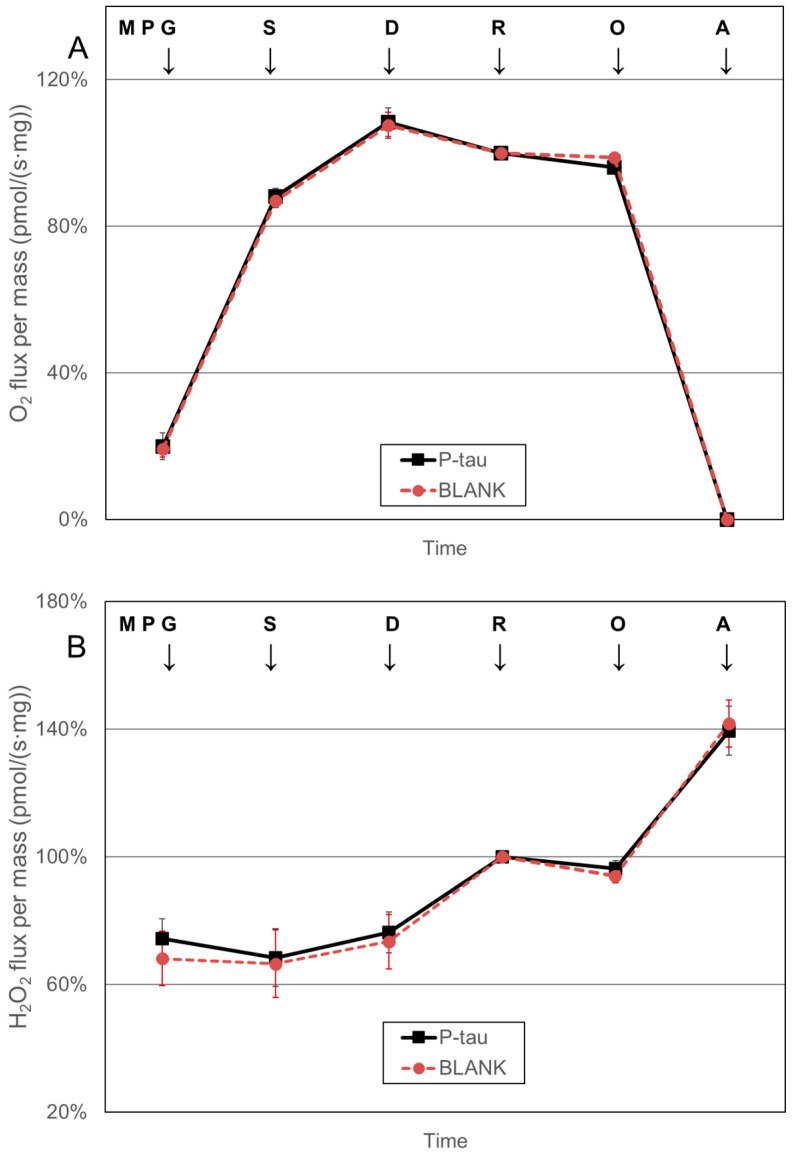
In vitro effect of the phosphorylated tau (P-tau, 60 nmol/L) on (**A**) respiratory rate and (**B**) hydrogen peroxide production (H_2_O_2_) in isolated brain mitochondria in defined coupling control states achieved in the complex I&II-linked electron transfer pathway. The kinetics of oxygen consumption and H_2_O_2_ production were measured simultaneously in respiratory states achieved after the addition of 2 mmol/L malate (M), 5 mmol/L pyruvate (P), 10 mmol/L glutamate (G), 10 mmol/L succinate (S), 1.25 mmol/L ADP (D), 1 µmol/L rotenone (R), 20 ng/mL oligomycin (O), and 1.25 µg/mL antimycin A to isolated brain mitochondria. Mitochondrial respiration is corrected for residual oxygen consumption (value after treatment with antimycin A) as the baseline state. The effect of P-tau and the blank solution (BLANK) is shown. The kinetics of mitochondrial oxygen consumption in defined respiratory states were normalized to the capacity of complex II-linked respiration (capacity of electron transfer system after complex I inhibition by rotenone); the rate of hydrogen peroxide production was normalized to the rate of H_2_O_2_ formation in this respiratory state. Values represent the mean ± SD of four independent experiments.

## Data Availability

The original data presented in the study are openly available in repository ZENODO at DOI 10.5281/zenodo.14794021 (https://zenodo.org/records/14794021, accessed on 18 March 2025).

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
