# Peer review of "Functional Analysis of Direct In Vitro Effect of Phosphorylated Tau on Mitochondrial Respiration and Hydrogen Peroxide Production"

_biomolecules, 2025, doi:10.3390/biom15040495_

Round 1
Reviewer 1 Report
Comments and Suggestions for Authors
To uncover the mechanisms leading to phosphorylated tau (P-tau)-induced mitochondrial toxicity, this study investigated the direct effects of P-tau on the kinetics of oxygen consumption and the associated H2O2 production in defined mitochondrial respiratory states in a biological model of isolated brain mitochondria in vitro. The results showed that P-tau at submicromolar concentrations had no significant effects on either mitochondrial respiration or hydrogen peroxide production in different respiratory states. There are some concerns about the present manuscript including typos, inconsistent writing type, and others, as listed in the following:
L169: (TCEP) v NaP, 0.78 mmol/L N-ethylmaleimide (NEM) v NaP
L171: 37.5 µmol/L TCEP a 30 µmol/L NEM in saline with TRIS
L207: (nmol O2 per sec
L215: H2O2 production
*L361-364: Authors should discuss the results and how they can be interpreted from the perspective of previous studies and of the working hypotheses. The findings and their implications should be discussed in the broadest context possible. Future research directions may also be highlighted??
**L401: References: it is better to keep consistent writing for for
(1) title: R1,2,4.. (capital prefix only on the first word) vs. R3,7,14…(capital prefix on all words)
(2) no page number: R3,6,45,61,82,83 (Cell Mol Neurobiol. 2023 Apr;43(3):951-961), R84,87,
Author Response
Response to Rewiever 1 comments and suggestions:
Thank you for pointing out the typos. All typos have been corrected:
Comments 1: L169: (TCEP) v NaP, 0.78 mmol/L N-ethylmaleimide (NEM) v NaP
Response 1: „v“ corrected to „in“, i.e. „(TCEP) in NaP, 0.78 mmol/L N-ethylmaleimide (NEM) in NaP“
Comments 2: L171: 37.5 µmol/L TCEP a 30 µmol/L NEM in saline with TRIS
Response 2: „a“ corrected to „and“, i.e. „37.5 µmol/L TCEP and 30 µmol/L NEM in saline with TRIS“
Comments 3: L207: (nmol O2 per sec
Response 3: „O2“ corrected to „O2“, i.e. „(nmol O2 per sec“
Comments 4: L215: H2O2 production
Response 4: „H2O2“ corrected to „H2O2“
Comments 5: *L361-364: Authors should discuss the results and how they can be interpreted from the perspective of previous studies and of the working hypotheses. The findings and their implications should be discussed in the broadest context possible. Future research directions may also be highlighted??
Response 5: This paragraph was deleted because it was an omission to delete text in the template for writing publications in Biomolecules.
Comments 6: **L401: References: it is better to keep consistent writing for for
(1) title: R1,2,4.. (capital prefix only on the first word) vs. R3,7,14…(capital prefix on all words)
(2) no page number: R3,6,45,61,82,83 (Cell Mol Neurobiol. 2023 Apr;43(3):951-961), R84,87,
Response 6: References have been corrected: (1) capital prefix is ​​used only in the first word; (2) page numbers are added.
Reviewer 2 Report
Comments and Suggestions for Authors
The authors have designed an appropriate combination of specific substrates and inhibitors of the phosphorylation pathway and enabled the measurement and functional analysis of the effect of P-tau on mitochondrial respiration in defined coupling control states achieved in complex I-, II-, and I&II-linked electron transfer pathways. At submicromolar P-tau concentrations, they found no significant effect of P-tau on either mitochondrial respiration or hydrogen peroxide production in different respiratory states. Titration of P-tau showed a nonsignificant dose dependent decrease in hydrogen peroxide production for complex I- and I&II-linked pathways. Little direct in vitro effect of P-tau oligomers on both mitochondrial respiration and hydrogen peroxide production indicates that P-tauinduced mitochondrial dysfunction in AD is not due to direct effects of P-tau on the efficiency of the electron transport chain and on the production of reactive oxygen species. I have the following comments:
- In abstract, what do the authors mean by the term 'Little direct effect"
- The introduction is too vast with many subsections. Reduce the introduction to a maximum of two pages. The introduction of an original research article should be concise and well organised with a proper flow. In introduction section I found some following issues that needs to be addressed:
References 10-13 are too old. If possible kindly replace.
The following sentences are redundant and similar sentences are repeated many times in the text. They do not add any new information. Please remove a few.:
"Hyperphosphorylation of tau leads to its dissociation from microtubules, which destabilizes them, disrupts axonal transport and delivery of synaptic proteins and organelles to the synapse, and contributes to neurodegeneration."
"P-tau aggregates are thought to be involved in the disruption of synaptic and mitochondrial function"
"The neurotoxicity of both Aβ and tau oligomers is realized and potentiated by mitochondrial dysfunction."
"The toxic effects of tau could also be realized through tau-induced mitochondrial membrane perturbation"
- Materials and methods:
Please abbreviate pH as 7.5 not 7,5.
All the stock solutions are prepared in tris buffer with glycerol as protectant. Mention the final concentration of glycerol in these solutions.
The authors have stated that "Mitochondria were isolated from the cerebral cortex of pigs". Please mention the animal ethical clearance number obtained to conduct these experiments.
Line # 221: "For CI-linked pathway activation and the effect of different concentrations of P-tau, further gradually add with 2 mmol/L malate.....". Please check the grammar.
The authors have stated that "Purified mitochondria were stored on ice in preservation medium...". Does a frozen mitochondria retains its proper functions like ROS production or H2O2 production? Outside a cell or body, does the mitochondria work in absence of its niche? Please clarify in section 2.2.
I recommend a major revision.
Author Response
Comments 1: In abstract, what do the authors mean by the term 'Little direct effect"
Response 1: “Little direct in vitro effect” has been replaced by “Nonsignificant in vitro effect”.
Comments 2: The introduction is too vast with many subsections. Reduce the introduction to a maximum of two pages. The introduction of an original research article should be concise and well organised with a proper flow. In introduction section I found some following issues that needs to be addressed:
References 10-13 are too old. If possible kindly replace.
Response 2: References 10-13 have been replaced with newer ones. Introduction has been reduced and edited.
Comments 3: The following sentences are redundant and similar sentences are repeated many times in the text. They do not add any new information. Please remove a few.:
"Hyperphosphorylation of tau leads to its dissociation from microtubules, which destabilizes them, disrupts axonal transport and delivery of synaptic proteins and organelles to the synapse, and contributes to neurodegeneration."
"P-tau aggregates are thought to be involved in the disruption of synaptic and mitochondrial function"
"The neurotoxicity of both Aβ and tau oligomers is realized and potentiated by mitochondrial dysfunction."
"The toxic effects of tau could also be realized through tau-induced mitochondrial membrane perturbation"
Response 3: The sentences are removed.
Comments 4: Materials and methods: Please abbreviate pH as 7.5 not 7,5.
Response 4: Corrected.
Comments 5: All the stock solutions are prepared in tris buffer with glycerol as protectant. Mention the final concentration of glycerol in these solutions.
Response 5: Glycerol as a protectant was only used in the supplied P-tau stock solution. Information in Materials and Methods has been corrected: “Chemicals were purchased from Merck KGaA (Darmstadt, Germany), except for (1) Amplex™ UltraRed from Molecular Probes (Eugene, OR 97402, USA) and (2) P-tau (TAU-H5147 Human Tau-441 GSK-3beta-phosphorylated protein) from ACROBiosystems Group (Newark, DE 19711, USA); P-tau was supplied and shipped with dry ice in buffer 50 mmol/L Tris, 150 mmol/L NaCl, pH 7.5, with glycerol as a protectant.”
Comments 6: The authors have stated that "Mitochondria were isolated from the cerebral cortex of pigs". Please mention the animal ethical clearance number obtained to conduct these experiments.
Response 6: Bioethics committee approval was not required for our in vitro experiments with pig brain mitochondria, because they were obtained from slaughter animals and no living animals were used during these experiments. The following information has been added to the beginning of chapter 2.2. Mitochondria isolation: “Mitochondria were isolated from the cerebral cortex of pig brain by a previously described method [63]. Pig brains were obtained from a local slaughterhouse within 1 hour after CO2 stunning of animals and killing them bleeding to death. Briefly, the brain cortex was gently homogenized in ten volumes (w/v) of ice-cold buffered sucrose 0.32 M supplemented with protease inhibitor cocktail.”
Comments 7: Line # 221: "For CI-linked pathway activation and the effect of different concentrations of P-tau, further gradually add with 2 mmol/L malate.....". Please check the grammar.
Response 7: Grammar has been corrected in all 4 paragraphs.
Comments 8: The authors have stated that "Purified mitochondria were stored on ice in preservation medium...". Does a frozen mitochondria retains its proper functions like ROS production or H2O2 production? Outside a cell or body, does the mitochondria work in absence of its niche? Please clarify in section 2.2.
Response 8: We apologize for the inappropriate description. Mitochondria were not frozen; they were stored at 2°C. The preservation of mitochondrial function with this method of isolation and preservation of mitochondria has been verified in previous experiments. E.g., the function of the mitochondrial respiratory chain is preserved with this method of isolation of mitochondria for at least two days (citation 67). In section 2.2. a correction was made: “Purified mitochondria were stored at 2°C (in an ice-cold water bath) in preservation medium at a protein concentration of approximately 40 mg/mL determined by the Lowry method.”
Comments 9: I recommend a major revision.
Response 9: The revision was made based on the comments of the reviewers.
Round 2
Reviewer 2 Report
Comments and Suggestions for Authors
The authors have addressed all the queries raised by the reviewers and now the manuscript can be accepted for publication.